# Roles of NRF3 in the Hallmarks of Cancer: Proteasomal Inactivation of Tumor Suppressors

**DOI:** 10.3390/cancers12092681

**Published:** 2020-09-20

**Authors:** Akira Kobayashi

**Affiliations:** 1Laboratory for Genetic Code, Graduate School of Life and Medical Sciences, Doshisha University, Kyotanabe, Kyoto 610-0394, Japan; akobayas@mail.doshisha.ac.jp; Tel.: +81-774-65-6273; 2Department of Medical Life Systems, Faculty of Life and Medical Sciences, Doshisha University, Kyotanabe, Kyoto 610-0394, Japan

**Keywords:** cancer, malignancy, proteasome, transcription, NRF3, NRF2, p53, Rb, anticancer drug

## Abstract

**Simple Summary:**

This review summarizes recent advances in our understanding of the physiological roles of the NFE2-related factor 2 (NRF2)-related transcription factor NRF3 in cancer. NRF3 confers cells with six so-called “hallmarks of cancer” through upregulating gene expression of specific target genes, leading to tumorigenesis and cancer malignancy. These driver gene-like functions of NRF3 in cancer are distinct from those of NRF2.

**Abstract:**

The physiological roles of the NRF2-related transcription factor NRF3 (NFE2L3) have remained unknown for decades. The remarkable development of human cancer genome databases has led to strong suggestions that NRF3 has functional significance in cancer; specifically, high *NRF3* mRNA levels are induced in many cancer types, such as colorectal cancer and pancreatic adenocarcinoma, and are associated with poor prognosis. On the basis of this information, the involvement of NRF3 in tumorigenesis and cancer malignancy has been recently proposed. NRF3 confers cancer cells with selective growth advantages by enhancing 20S proteasome assembly through induction of the chaperone gene proteasome maturation protein (*POMP*) and consequently promoting degradation of the tumor suppressors p53 and retinoblastoma (Rb) in a ubiquitin-independent manner. This new finding offers insight into the proteasomal but not the genetic inactivation mechanism of tumor suppressors. Moreover, NRF3 promotes cancer malignancy-related processes, including metastasis and angiogenesis. Finally, the molecular mechanisms underlying NRF3 activation have been elucidated, and this knowledge is expected to provide many insights that are useful for the development of anticancer drugs that attenuate NRF3 transcriptional activity. Collectively, the evidence indicates that NRF3 confers cells with six so-called “hallmarks of cancer”, implying that it exhibits cancer driver gene-like function. This review describes recent research advances regarding the newly discovered addiction of cancer cells to NRF3 compared to NRF2.

## 1. Introduction

In the past decade, remarkable advances in human cancer databases, including genome, transcriptome, proteome, epigenome, metabolome, and clinical data, have led to many insights into the biology of cancer and have suggested new concepts, such as driver and passenger gene mutations [1,2,3,4]. Driver gene mutations confer tumor cells with selective growth advantages, and more than 200 driver genes, such as *PIK3CA*, *SMAD4,* and *TP53,* have been identified. Passenger gene mutations are merely “passengers” and have no effect on tumorigenesis. Cancer arises through a multistep mutagenic process whereby cancer cells acquire a common set of properties, named “the hallmarks of cancer”, originally proposed by Hanahan and Weinberg [1]. These hallmarks include sustained proliferative signaling, evasion of growth suppressors, resistance to cell death, enabling of replicative immortality, induction of angiogenesis, activation of invasion and metastasis, reprogramming of energy metabolism, and evading immune destruction. Luo et al. [5] proposed additions to the cancer hallmark list: activation of stress responses to proteotoxic stress, metabolic stress, mitotic stress, and DNA damage stress.

Approximately 20 years ago, the transcription factor NFE2-related factor 3 (NRF3, NFE2L3) was discovered by my group as the sixth and final cap ‘n’ collar (CNC)-basic leucine zipper (bZip) transcription factor family comprising p45/NFE2, NRF1 (NFE2L1), NRF2 (NFE2L2), BACH1, and BACH2 [6,7,8]. The physiological function of NRF3 remained a mystery for years, in part because *Nrf3*-deficient mice exhibited no apparent abnormalities under physiological conditions [9,10,11]. Nevertheless, even at that time, NRF3 was considered to be involved in the biology of cancer because of its extremely high expression levels in human colon cancer (Figure 1) [12]. Furthermore, current databases indicate an association between NRF3 expression and poor prognosis, suggesting an essential roles of NRF3 in cancer malignancy [13]. For this reason, NRF3 has finally attracted attention, and the number of studies reporting intriguing functions of NRF3 in cancer is gradually increasing. Recent findings show that NRF3 confers cells with selective growth advantages, namely, the six hallmarks of cancer (Figure 2), implying that NRF3 also exhibits cancer driver gene-like functions. The functions of NRF3 are distinct from those of the NRF3 homolog NRF2 that has been recognized as a cancer driver gene [7,14,15].

This review summarizes the current advances in our knowledge regarding the physiological roles and regulatory mechanisms of NRF3 in cancer, especially its regulation by the proteasomes, compared to those of NRF2. The expected development of cancer therapeutic interventions for this axis is also discussed.

## 2. Newly Discovered Addiction of Colon Cancer Cells to NRF3

### 2.1. Remarkable NRF3 Induction Associated with Malignancy

*NRF3* mRNA expression is abundant in the cornea, skin, bladder, and placenta, but it is low, albeit ubiquitous, in other tissues. Importantly, The Cancer Genome Atlas (TCGA) indicates that the *NRF3* gene is remarkably upregulated in many cancer tissues, such as colorectal adenocarcinoma (COAD), rectal adenocarcinoma (READ), and pancreatic ductal adenocarcinoma (PDAC; it is annotated as PAAD within TCGA) (Figure 1). A comparative analysis of 921 gastrointestinal adenocarcinomas has also shown that *NRF3* expression is upregulated in esophageal adenocarcinoma (EAC) and has confirmed its upregulation in COAD and READ, with *NRF3* expression in READ being correlated with that of stemness markers [17]. Furthermore, correlation analyses of DNA methylation in clear cell renal cell carcinoma (ccRCC) have revealed that hypomethylation of the *NRF3* gene leads to elevated expression levels and poor survival [18,19]. Supporting biological links between NRF3 and cancer malignancy, high *NRF3* induction is correlated with poor prognosis in PDAC (Figure 1). Finally, *NRF3* has been reported to be one of 127 significantly mutated genes among 12 cancer types [20].

We investigated the impacts of *NRF3* induction on cancer cell growth and resistance to cell cycle arrest and found that *NRF3* knockdown significantly reduced cell proliferation in a p53-dependent manner [21]. Bury et al. [22] have also reported NRF3-mediated tumorigenesis. They identified that upregulation of NRF3 in human colon cancer specimens was correlated with poor prognosis. Consistent with our results, silencing NRF3 significantly inhibited colon cancer cell proliferation in vitro and in vivo. This accumulating evidence strongly highlights the possibility that NRF3 drives tumorigenesis and cancer malignancy.

### 2.2. Mechanisms of the NRF3 Induction in Colon Cancer

With respect to the molecular basis underlying *NRF3* induction in colon cancer cells, research has revealed two distinct molecular pathways (Figure 3A). First, we discovered that the β-catenin/T-cell specific transcription factor (TCF4) complex in the Wnt signaling pathway induces *NRF3* gene expression in several types of cancer cells, including colon cancer cells [12]. The Wnt/β-catenin pathway, which is essential for normal intestinal growth and development, plays a vital role in the initiation of colorectal cancer progression [23,24,25]. Mutation of the adenomatous polyposis coli (*APC*) gene and the *CTNNB1* gene encoding β-catenin is an initial event driving to the development of adenoma in most sporadic cases [26]. Constitutive activation of β-catenin due to *APC* mutations causes hyperplastic epithelium. The β-catenin/TCF4–NRF3 axis was confirmed through studies on intestinal epithelial organoids derived from *Apc*-null mice, suggesting the species conservation of this gene regulatory mechanism. Second, Bury et al. [22] found that the NF-κB complex, including RELA, regulates *NRF3* expression in colon cancers. While they discovered the mechanism by analyzing the same human colon cancer HCT116 cells that we analyzed, it remains unclear as to whether these two distinct regulatory mechanisms of *NRF3* gene expression in colon cancer cells are cooperative or mutually exclusive.

### 2.3. NRF3 Transcriptome in Cancer Cells

Similar to other CNC family proteins, NRF3 also augments gene expression by binding to the antioxidant response elements (AREs) in target genes via heterodimerization with small Maf proteins (sMafs) [13,31]. NRF2 is known to regulate cytoprotective gene expression counteracting oxidative stress, and NRF1 sustains protein homeostasis (i.e., proteostasis) by mediating proteasome gene expression upon proteasome inhibition. Nevertheless, the NRF3 target genes remained elusive. To understand the physiological roles of NRF3 in cancer, genome-wide transcriptome analyses, including chromatin-immunoprecipitation (ChIP) sequencing analyses, are indispensable. In this regard, Chénais et al. [32] conducted a transcriptome analysis using HCT116 cells treated with tumor necrosis factor α (TNFα), which induces NRF3 expression, and identified the double homeobox 4 (DUX4), a repressor of cyclin-dependent kinase 1 (CDK1), as one of the NRF3 target genes [22]. Liu et al. [33] reported the results of ChIP-exo sequencing of NRF1, NRF2, and NRF3 (NRF1–NRF3) using U2OS cells stably overexpressing these proteins. They categorized all ARE-containing genes into six groups according to their cooperative or independent regulation by NRF1–NRF3 and found that NRF3 directly regulates the expression of genes related to cell proliferation and metabolism. Because of the overexpression experiment used, this conclusion needs to be further confirmed under physiological conditions.

We also performed microarray and ChIP-sequencing experiments using HCT116 cells treated with the proteasome inhibitor MG132, which also stabilizes the NRF3 proteins [34]; however, unfortunately, we could not draw a definitive conclusion due to the small number of ChIP peaks. Despite the situation, our data indicate that NRF3 regulates the expression of proteasome-related genes, i.e., there is a biological relationship between NRF3 and proteostasis in cancer.

### 2.4. Cancer Cells are Highly Addicted to Proteostasis

The ubiquitin–proteasome system (UPS), the primary mechanism by which proteins are degraded in the cytoplasm and nucleus, determines cellular protein levels and quality in coordination with autophagy and chaperon-mediated protein folding [29,30,35,36]. Cancer cells express massive amounts of both wild-type and mutant proteins to sustain their growth and high metabolic rate. This cellular situation in turn may cause the accumulation of overproduced or misfolded proteins; cancer cells therefore profoundly depend on (or are addicted to) high content and activity levels of proteasomes, which are responsible for protein homeostasis (proteostasis) or protein quality control. This hallmark of cancer is its “Achilles’ heel”, as cancer cells are more susceptible than normal cells to proteasome inhibition [5]. Bortezomib, which utilizes this Achilles’ heel of cancer cells, was the first proteasome inhibitor applied in the clinic for the treatment of multiple myeloma and is currently utilized for mantle cell lymphoma therapy [35].

### 2.5. Gene Regulation and Assembly Mechanisms of the Proteasomes

The 26S proteasome is a major proteasome in cells that rapidly degrades polyubiquitinated proteins. It consists of the 20S proteasome and 19S regulatory particles; the former degrades substrate proteins, and the latter recognizes and unfolds ubiquitinated proteins in an ATP-dependent manner, enabling proteolysis via the 20S proteasome (Figure 3B). The 26S proteasome is an unusually large complex that consists of 66 subunits encoded by 33 genes on individual chromosomes. In yeast, the regulatory mechanisms of both constitutive and inducible 26S proteasome expression are modulated by the transcription factor Rpn1. In higher eukaryotes, transcription factors of 26S proteasome remained elusive for a long time; specific transcription factors have only recently been reported to mediate gene expression [37]. For example, NRF2 and NRF1 directly activate the gene expression of 26S proteasome subunits by binding to ARE sites in their regulatory regions. NRF2 eliminates the accumulation of damaged proteins under oxidative stress conditions [38]. NRF1 rescues 26S proteasome function by upregulating gene expression in response to proteasome dysfunction and inhibition [39,40,41,42]. This cellular response is called the “proteasome bounce-back response” (or proteasome recovery), and it is crucial for maintaining proteostasis under conditions of proteasome dysfunction.

Following the coordinated induction of 33 proteasome subunit genes, these subunits are tightly assembled via complex mechanisms [28,29,30,36]. The 20S proteasome includes an α-ring and a β-ring that contain α1–α7 and β1–β7 subunits, respectively. These rings are stacked in an α-β-β-α topology to form the cylindrical 20S proteasome. α-ring assembly is modulated by proteasome assembling chaperone 1–4 (PAC1–PAC4), and β-ring assembly, half-proteasome dimerization, and 20S proteasome maturation are orchestrated by the proteasome maturation protein (POMP) chaperone. Intriguingly, high *POMP* expression is associated with poor prognosis in endoplasmic reticulum (ER) α-positive breast cancers, and *POMP* expression is mitigated by the tumor suppressor miR-101 in normal cells [43]. POMP expression is modulated by NRF3 in colon cancer cells, as described in the next section [21]. The 19S proteasome includes a base and lid: the base is composed of six AAA^+^-ATPase subunits (Rpt1–Rpt6) and four non-ATPase subunits (Rpn1, Rpn2, Rpn10, and Rpn13), and the lid is composed of nine non-ATPase subunits (Rpn3, Rpn5–Rpn9, Rpn11, Rpn12, and Sem1). Five chaperones also regulate the assembly of the base and lid. Aside from the 19S proteasome, other proteasome regulators, such as proteasome activator (PA) 28αβ, PA28γ, PA200, ECP29, and PSMF1, are also known to bind to the 20S proteasome, forming alternative proteasome complexes that modulate divergent types of proteasomal degradation [29,30,44].

### 2.6. NRF3 Stimulates 20S Proteasome Assembly and the Cell Cycle

Two physiological roles of NRF3 in colon cancer cells have been found (Figure 2 and Figure 3A). NRF3 augments the gene expression of POMP as a molecular chaperone that is essential for the assembly of the 20S proteasome [21]. Although the biological function of the 20S proteasome itself remains unclear, we found that 20S proteasome activation by the NRF3–POMP axis leads to degradation of the tumor suppressor proteins p53 and retinoblastoma (Rb). More than 50% of solid tumors harbor *TP53* gene mutations [45,46]. In colorectal cancers, it is well known that the *TP53* gene is ultimately mutated, which occurs after mutation of *APC*, hypermethylation of tumor suppressor genes, and KRAS activation during carcinogenesis (Figure 4) [26]. Thus, the NRF3–POMP axis-activated assembly of the 20S proteasome is likely indispensable to tumorigenesis because it should inhibit the tumor suppressor function of p53 at the posttranslational level, before its gene is mutated. We further discovered the glucose transporter 1 (GLUT1) as one of the NRF3 target genes, although whether the NRF3–GLUT1 axis activates glycolysis is unresolved (Figure 2) [12]. Our findings reveal the new molecular mechanism underlying p53 suppression in tumorigenesis, as discussed in Section 2.8.

Next, Bury et al. [22] also reported on NRF3-mediated tumorigenesis. They found that upregulation of *NRF3* in human colon cancer specimens correlated with poor prognosis. Consistent with our results, silencing of NRF3 significantly inhibited colon cancer cell proliferation in vitro and in vivo. The molecular basis of tumorigenesis is the NRF3-mediated downregulation of DUX4, a repressor of CDK1, although how NRF3 functions as a transcriptional repressor is unresolved. Collectively, these insights may suggest the presence of multiple pathways of NRF3 in colon cancer growth.

### 2.7. Colon Cancer Cells May Favor NRF3 for Proteostasiss

#### NRF3 Suppresses NRF1 Translation by Inducing CPEB3

While investigating the biological relationship between NRF3 and proteostasis in cancer, we discovered one more regulatory mechanism—cross talk between NRF3 and NRF1 in colon cancer cells (Figure 3A) [27]. NRF3 attenuates NRF1 expression at the protein level but not at the mRNA level in HCT116 cells. In the underlying molecular mechanism, NRF3 directly augments the expression of the translational regulator cytoplasmic polyadenylation element-binding protein 3 (CPEB3) by binding to the ARE site in the regulatory region. CPEB3, which belongs to the RNA-binding CPEB family, both positively and negatively regulates expression at the posttranscriptional level via polyadenylation of the 3′-untranslated region (UTR) and ribosome recruitment onto mRNAs [47,48,49]. Indeed, CPEB3 reduces polysome formation on *NRF1* mRNA by directly binding to the CPE site on its 3’UTR. This observation indicates functional differences between NRF3 and NRF1, namely, that cancer cells favor NRF3 transcriptional pathways over NRF1 pathways. In addition, this finding implies the functional similarity between these proteins and the importance of their transcription because silencing of *NRF3* immediately induces an increase in NRF1 protein levels by eliminating CPEB3-mediated translational repression, thereby rescuing the loss of NRF3 function in cells. Further extensive genome-wide transcriptome analyses involving RNA sequencing and ChIP-sequencing should be conducted to elucidate this process.

Moreover, the finding also raises the possibility that NRF3 enhances tumorigenesis and malignant transformation by regulating the translation of cancer-related genes via induction of *CPEB3* gene expression, although the whole set of CPEB3 target genes remains uncharacterized. Curiously, the *CPEB3* gene has been reported to be downregulated in several cancer tissues, including colorectal cancer [48,49,50]. Since human colorectal cancer is classified into subtypes with distinct cellular characteristics [51], the NRF3–CPEB3 axis might be responsible for tumorigenesis of a particular subtype of colorectal cancer. Notably, our findings add CPEB3 to the list of ARE-containing genes whose expression is regulated by CNC family proteins. Accordingly, NRF2 and NRF1 may also exert some biological functions through translational regulation of the expression of some genes via *CPEB3* induction.

### 2.8. Conceptual Hypotheses: The Oncogenic Stress Response Hypothesis and the Proteasomal Inactivation of Tumor Suppressors Hypothesis

What new concepts can be derived from these discoveries about NRF3-mediated tumorigenesis? Our understanding of NRF3 is still early in its development; therefore, many experiments, including in vivo analyses, have not yet been conducted. However, two conceptual hypotheses are proposed for future research: the oncogenic stress response hypothesis and the proteasomal inactivation of the tumor suppressors hypothesis (Figure 4).

Oncogenic stress refers to the activation of an individual oncogene in normal cells. Oncogenic stress paradoxically induces permanent cell cycle arrest or senescence as a tumor-suppressive response [52,53,54]. For instance, activation of oncogenes such as β-catenin, MYC, and RAS activates the tumor suppressor ARF, resulting in the degradation of MDM2 and thereby the stabilization of p53 [55,56,57,58]. During colorectal cancer development, mutation of the *APC* or *β-catenin* genes in the Wnt signaling pathway is the first crucial molecular event [26], and such mutation causes oncogenic stress. Then, β-catenin promotes the stabilization of p53 by diminishing the MDM2 function via the ARF induction [59]. Because the *TP53* mutation occurs at the final step in tumorigenesis, the oncogenic stress-driven tumor suppressor function of p53 must be inhibited during prior steps. Moreover, oncogenic stress also mitigates the function of the cyclinD1/CDK4 complex by upregulating the splicing variant p16, consequently activating the Rb protein [58]. It is possible that the NRF3–POMP–20S proteasome axis is required for counteracting oncogenic stress during tumorigenesis.

In the classical Knudson two-hit model, tumor development requires a two-step process in which both alleles of a tumor suppressor gene are silenced because the tumor suppressor gene is recessive; however, mutation of tumor suppressor genes has been recently found to result in haploinsufficiency as well as gain-of-function and dominant-negative phenotypes [60]. In this regard, the NRF3–POMP–20S proteasome axis can suppress the tumor-suppressive function of p53 and Rb at the protein level, and it may efficiently inactivate recessive tumor suppressor genes without the second hit (or perhaps even without the first hit). Confirmation of these two attractive hypotheses would enable the conceptual generalization of the biological function of NRF3 in cancer.

### 2.9. Remaining Issues Regarding the NRF3–POMP–20S Proteasome Axis

Our findings on the NRF3–POMP–20S proteasome axis have successfully revealed the physiological function of NRF3 in tumorigenesis (Figure 3); however, we admit that our data do not directly prove that the 20S proteasome itself, which is activated by the NRF3–POMP axis, carries out the p53 or Rb degradation. The biological function of the 20S proteasome is still unclear and controversial because of the scarce literature [61,62]. It has been reported that the 20S proteasome degrades proteins that contain partially unfolded regions, such as intrinsically disordered and oxidatively damaged proteins; these proteins are capable of entering the catalytic chamber without being unfolded by the 19S proteasome. As intrinsic disordered proteins, p53 and Rb are likely degraded by the 20S proteasome in a ubiquitin-independent fashion. Though we observed the 20S proteasome in the 20S fraction of whole-cell extracts in our glycerol density gradient experiments, we cannot rule out the possibility that our “20S proteasome” contained some accessory or regulatory particles along with the 20S proteasome. Alternative proteasomes comprising the 20S proteasome and the proteasome regulators PA28αβ, PA28γ, PA200, ECP29, and PSMF1 contribute to the diversity of proteasomal degradation [30]. These proteasome regulators may modulate substrate specificity and ubiquitin-independent degradation.

The biological relationship between the 20S proteasome and the ubiquitin ligase MDM2 in p53 degradation is also unclear. Given that MDM2 degrades p53 in a ubiquitin–proteasome-dependent fashion under physiological conditions [63], the 20S proteasome pathway appears unnecessary. Hence, we surmise that MDM2-mediated p53 degradation is suppressed upon oncogenic stress as described in Section 2.8; consequently, the 20S proteasome is required for the repression of the p53-driven cell growth arrest and apoptosis that occur in tumorigenesis. Moreover, it is unclear whether p53 mutants are degraded by the NRF3-activated 20S proteasome. p53 mutants gain function during carcinogenesis, despite failing to bind to DNA [45,46,64]. For instance, p53 mutants have been found to cooperate with NRF2 to activate proteasome gene transcription, resulting in resistance to the proteasome inhibitor carfilzomib [65].

Finally, the ARE sequence in the *POMP* gene is slightly different from the consensus sequence—the ARE sequence in the *POMP* gene is TGAGNNNCG, and the consensus ARE sequence is TGA(G/C)NNNGC. The underlined 3′-GC sequence in the ARE site includes crucial nucleotides recognized by sMaf proteins, which dimerize with CNC proteins [7]. This difference may imply that the partner molecules of NRF3 are not sMaf proteins. Identification of the heterodimerizing partner molecules, the DNA recognition sequence, and target genes of NRF3 is critical.

## 3. NRF3-Driven Cancer Malignancy

Emerging evidence has revealed that NRF3 also confers tumor cells with malignancy; specifically, it activates the invasion and metastasis of thyroid cancer cells (Figure 2) [66]. NRF3 has been identified as a pivotal downstream effector of “regulator of calcineurin 1, isoform 4” (RCAN1-4), which suppresses endothelial cell migration, neovascularization, and tumor growth. Overexpression of NRF3 enhances both RCAN1-4-dependent and RCAN1-4-independent cell invasion. Among human clinical samples, distant metastatic samples from thyroid cancer patients exhibit upregulation of *NRF3* gene expression. *NRF3* knockdown also substantially suppressed the migration, invasion, and epithelial–mesenchymal transition (EMT) of hepatocellular carcinoma cells [67]. Supporting these reports, we also observed that NRF3 overexpression in human H1299 lung cancer cells led to enhanced metastasis from the spleen to the liver as well as tumor growth in a xenograft model [21]. Moreover, a biological relationship between NRF3 and angiogenesis in pancreatic adenocarcinoma has been reported [68]. It seems that vascular endothelial growth factor A (*VEGFA)* induction is indirectly regulated by NRF3 because of the absence of ARE sites in the *VEGFA* gene, but the details of the molecular mechanisms remain to be discovered.

## 4. NRF3 as the Tumor Suppressor

Interestingly, it has also been reported that NRF3 may be involved in opposing tumor suppression. In gastric and breast cancer, the expression of *NRF3* is negatively correlated with DNA methylation, and hypermethylation of the *NRF3* gene is associated with shorter overall survival [69,70]. Indeed, NRF3 suppresses breast cancer cell metastasis and cell proliferation, implying that NRF3 is a favorable predictor of survival in breast cancer [71]. In addition, NRF3 is expressed at low levels in colorectal cancer, and its downregulation promotes colorectal cancer malignancy [72]. On the basis of the Gene Expression Omnibus (GEO) and TCGA datasets, research has identified NRF3 as a potential diagnostic and prognostic biomarker gene for colorectal cancer [73]. The results of the latter two reports contradict the results regarding the involvement of NRF3 in tumorigenesis. The reason for the discrepancies among these results is still unknown, but NRF3 may coordinate carcinogenesis either positively or negatively, depending on the cell type and context or stage of carcinogenesis. Further investigation is required to resolve this issue.

## 5. Differences between NRF3 and NRF2

### 5.1. Phenotypes of Gene Targeting Mice

Studies on *Nrf2*-knockout (KO) mice have provided great insights, revealing that Nrf2 plays indispensable roles in sustaining cellular homeostasis, including in the context of the oxidative stress response, as well as the fact that Nrf2 dysfunction leads to many diseases, including cancer [7]. Compared with wild-type mice, *Nrf3*-null mice display no apparent disorder under physiological conditions [9,10] and show no altered sensitivity to the phenolic antioxidant *tert*-butylhydroquinone (tBHQ) [11]. *Nrf3*-KO mice are reported to be highly susceptible to the environmental pollutant benzo[a]pyrene (B[a]P), developing T cell lymphoblastic lymphoma [74]. To determine any functional redundancy among CNC family proteins, researchers have generated *Nrf3*::*Nrf2* and *Nrf3*::*p45*-compound KO mice, but these mice do not show abnormalities beyond those of the *Nrf2*- and *p45*-KO mice [9,10]. These data strongly suggest a physiological function of Nrf3 that differs from the cytoprotective functions of Nrf2 against oxidative insults in vivo. The differences between these proteins are likely attributable to differences in their protein structures and expression patterns. Given our findings for NRF3 and NRF1, the loss of Nrf3 function is likely compensated by Nrf1.

### 5.2. Protein Structures and Regulatory Mechanisms

The molecular mechanisms underlying the oxidative stress response mediated by the KEAP1–NRF2 system mainly involve deregulation of KEAP1-mediated NRF2 degradation in response to stress [7,14]. Under physiological conditions, KEAP1 interacts with the Nrf2-ECH homology 2 (Neh2) domain of NRF2 and promotes its degradation by a Cul3-based E3 ubiquitin ligase (Figure 5A) [75,76,77,78]. KEAP1 senses oxidative insults via cysteine residues and subsequently hampers the ubiquitination activity of the Cul3 complex, activating NRF2 [79]. Thus, the Neh2 domain is a crucial interface for regulation by KEAP1.

The absence of the Neh2 domain in NRF3 implies that NRF3 function is not controlled by KEAP1. Rather, NRF3 has N-terminal homology box 1/2 (NHB1/2) domains [13,31,80,81]; the NHB1 domain sequesters NRF3 in the endoplasmic reticulum (ER), and the NHB2 domain contains the proteolysis site for the aspartic protease DNA damage-inducible 1 homolog 2 (DDI2). Under normal conditions, NRF3 is glycosylated and sequestered in the ER and is further degraded by the endoplasmic reticulum-associated degradation (ERAD) ubiquitin ligase HRD1 [34], which suppresses its function (Figure 5B). Accordingly, as in the case of oxidative stress in the NRF2 pathway, NRF3-activating stress and/or signals and their sensor molecules should exist. The fact that NRF3 functions in cancer cells means that NRF3-activating stress occurs in cancer cells. The activating signal is likely to induce the nuclear entry of NRF3 by stimulating DDI2-mediated cleavage of NRF3, thereby releasing NRF3 from repression mechanisms. We have determined that the DDI2 cleavage site is the AWLVH motif in the NHB2 domain of NRF3 [34]. Similar to the case for NRF1, this processing appears to require the deglycosylation of NRF3 by the enzyme NGLY1 [82,83], the product of the gene responsible for the *NGLY1*-deficiency disease, a congenital disorder [84]. Additionally, cleavage by DDI2 and the yeast homolog DNA damage-inducible protein 1 (DDI1) requires polyubiquitination of substrates [85,86]. Indeed, knockdown of *HRD1*, which is the E3 ubiquitin ligase of NRF3 in the ER, leads to the significant accumulation of NRF3 proteins in the cytoplasm.

In the nucleus, NRF3 augments the expression of genes such as *POMP* and *CPEB3*. NRF3 is ultimately degraded by the proteasome after ubiquitination by an E3 ubiquitin ligase with β-transducin repeat containing E3 ubiquitin protein ligase (β-TRCP) or F-box and WD repeat domain-containing 7 (FBW7) as substrate adaptors [34,87]. The nuclear degradation mechanism abrogates NRF3 activity after the stress disappears. The β-TRCP-mediated degradation system in the nucleus is shared by NRF2 [7]. Collectively, these insights clearly suggest the differences in molecular regulation between NRF3 and NRF2 and thereby explain their functional differences.

### 5.3. Missense Mutations in Cancer

TCGA database shows several missense mutations in the genes of *NRF2* and its regulatory factor *KEAP1* in many cancer tissues [7]. The mutated amino acids are located in the regulatory domains that are required for the crucial functions of these proteins, i.e., the domains that enable the interaction between NRF2 and KEAP1. These mutations constitutively activate NRF2, presumably leading to tumorigenesis. Meanwhile, some missense mutations in the *NRF3* gene are found in TCGA database, but these mutations do not seem to affect NRF3 function, on the basis of domain function analyses of NRF3. The lack of effect is plausible because NRF3 is expressed at low levels in normal cells; therefore, point mutations should not lead to tumorigenesis.

### 5.4. Functional Redundancy between NRF3 and NRF1

On the basis of the molecular evolution of the CNC family proteins as determined by their protein structures, *NRF3* is a rather close homolog of *NRF1* [88]. The protein structure of NRF3 is similar to that of NRF1; Nrf1 also possesses the NHB1/2 domain, and its transcriptional function is therefore similarly regulated by both DDI2 and NGLY (Figure 5A) [29,89,90]. *NRF1* is ubiquitously expressed in whole tissues. NRF1 and NRF3 complimentarily sustain basal proteasome activity in cancer cells [27], indicating a functional redundancy between NRF3 and NRF1, as described in Section 2.7. Nevertheless, *Nrf1*-deficient mice exhibit embryonic lethality due to fetal liver dysfunction and anemia [91], while *Nrf3*-null mice display no apparent abnormality. These results suggest the existence of functional differences between these proteins. Alternatively, these results may also indicate that Nrf1 can compensate the loss of Nrf3 function in mice. Intriguingly, Nrf1 carries the Neh2 domain as the KEAP1-regulatory interface, indicating the possibility that Keap1 controls Nrf1 function. While it has been reported that KEAP1 stabilizes the NRF1 protein [92], the detailed molecular mechanisms remain unknown.

## 6. Cross Talk of NRF3 with Other Transcription Factors

It is possible that communications among CNC factors support diversity and complexity among higher eukaryotes. Intriguingly, an indirect interaction between NRF2 and BACH1 has been recently reported to predispose patients to lung cancer metastasis [93,94]. We have also revealed NRF3 translationally represses NRF1 by inducing the *CPEB3* gene, which maintains basal proteasome activity in cancer (Figure 3A) [27]. Regarding its relationships with NRF2, we preliminarily discovered that NRF3 regulates *NRF2* expression in a certain cellular context. Moreover, NRF3 is likely to signal to other master transcription factors related to metabolism and EMT. We are currently examining the significance of these physiological processes in tumorigenesis and cancer malignancy and intend to publish the results in the near future.

## 7. Development of New Anticancer Drugs that Suppress NRF3

Given the essential roles of NRF3 in tumorigenesis and cancer malignancy, NRF3 inhibitors are expected to act as anticancer drugs by abrogating NRF3 function. To this end, elucidation of the NRF3 regulatory mechanisms [13,31] has suggested two strategies to suppress NRF3 biological functions in cancer cells (Figure 5B). The first strategy is to target the aspartic peptidase DDI2. As described in Section 5.2, DDI2 activates the nuclear translocation of NRF3 by processing its NHB2 domain, after which NRF3 target genes are expressed [34,95,96]. Accordingly, DDI2 inhibitors can suppress NRF3-driven tumorigenesis and cancer malignancy. DDI2 possesses a retroviral-like protease (RVP) domain whose structure is similar to that of the HIV-1 protease (HIVp), an HIV therapeutic target [97]. We are thus likely able to exploit HIV inhibitors as indirect NRF3 repressors. Consistently, the HIV inhibitors ritonavir, nelfinavir, and saquinavir have been found to attenuate proteasome activity and inhibit cancer cell growth in vitro [98,99]. The second strategy is to suppress the deglycosylation enzyme NGLY1 because deglycosylation by NGLY is required for DDI2 processing of NRF1 and the *Caenorhabditis elegans* (*C. elegans)* ortholog Skn-1A [82,83,100,101,102]. We observed that NGLY1 controls the nuclear translocation of NRF3. The biological linkage between NRF3 and NGLY1 may imply that NRF3 is involved in the pathological mechanism of congenital *NGLY1* deficiency disease [84].

Notably, DDI2 and NGLY1 inhibitors attenuate both NRF3 and NRF1 functions simultaneously because DDI2 and NGLY1 positively control the nuclear translocation mechanisms of these proteins. This finding implies that these compounds can improve the therapeutic effects of proteasome inhibitors such as bortezomib. As described above, cancer cells counteract the effects of proteasome inhibitor treatments by inducing proteasome gene expression (i.e., the proteasome bounce-back response), and both NRF1 and NRF3 mediate this response. Accordingly, cotreatment with either a DDI2 or an NGLY1 inhibitor should diminish the proteasome bounce-back response to proteasome inhibitor treatment, thereby increasing its efficacy. This idea appears promising because DDI2 inhibition by HIV protease inhibitors and NLGY1 inhibition by WRR139 significantly potentiate the effects of proteasome inhibition on the treatment of myeloma cells, such as multiple myeloma (MM) and T-cell acute lymphoblastic leukemia (T-ALL) cells [99,101].

## 8. An Emerging Concern: The Authenticity of Commercial Anti-NRF3 Antibodies

To support further development of NRF3 research, the authenticity of commercially available antibodies should be considered. Many companies present Western blot data, and these antibodies recognize NRF3 proteins of various molecular weights. We have recently succeeded in generating a mouse monoclonal anti-NRF3 antibody (#9408) that recognizes endogenous human NRF3 proteins of multiple molecular sizes, from approximately 100 kDa to 140 kDa, in whole-cell extracts of human cancer cells [34]. Because three independent siRNAs eliminate these bands completely, we believe that these immunopositive bands represent endogenous NRF3 proteins with distinct posttranslational modifications. Our anti-NRF3 antibody does not cross-react with the homologous proteins NRF1 and NRF2, but it unfortunately does not recognize the mouse Nrf3 protein, perhaps because of a low conservation of the epitope sequence (amino acids 364–415). The hybridoma against human NRF3 is available from the RIKEN BioResource Center, where it has been deposited (RCB4901, https://cellbank.brc.riken.jp/cell_bank/CellInfo/?cellNo=RCB4901&lang=En). To our knowledge, an antibody distributed by Sigma-Aldrich (St. Louis, MO, USA) (HPA055889) is also suitable for the Western blotting and immunostaining analyses of human NRF3. The authenticity of available antibodies is one of the crucial issues that must be addressed for further elucidation of the biological roles of NRF3 in carcinogenesis.

## 9. Conclusions and Perspectives

In the past decade, breakthroughs have finally been achieved in the NRF3 research field, and the physiological roles of NRF3 in tumorigenesis and cancer malignancy are gradually receiving attention worldwide. The available evidence strongly suggests that NRF3 confers cells with selective growth advantages, namely, the six hallmarks of cancer (Figure 2). Accordingly, NRF3 exhibits driver gene-like functions. We are currently testing a hypothesis suggesting that NRF3 is involved in another cancer hallmark, “avoidance of immune destruction”. However, compared with the remarkable research progress on NRF2, this breakthrough is only a small step in a long research journey. Much information about NRF3 must still be clarified, such as its transcriptome, activation signal, regulatory mechanisms, and the details of its biological and physiological roles. Indeed, we have previously found that NRF3 is involved in transcription related to many biological phenomena including glycolysis and EMT. This information may make it possible to utilize NRF3 for clinical applications to cancer therapy and diagnosis.

Clinical studies have strongly suggested a relationship of NRF3 with pancreatic cancer, given the remarkable induction of *NRF3* expression levels in PDAC cases with poor prognosis of (Figure 1) [13,16]. Additionally, NRF3 has been reported to promote cell migration and invasion and angiogenesis in PDAC [68]. Pancreatic cancer is one of the most lethal malignant diseases, and the survival rates of pancreatic cancer have remained relatively low. Overcoming pancreatic cancer is thus an urgent and vital issue worldwide. My laboratory is currently investigating whether NRF3 can be used as a therapeutic target to prevent PDAC malignancy or as an early diagnostic marker for pancreatic cancer.

An obstacle to the further development of NRF3 research is the lack of identification of an NRF3-activating stress signal. NRF3 must be an inducible transcription factor that responds to as-yet unidentified signals because its transcriptional activities are controlled by a subcellular localization mechanism, namely*,* the regulation of its nuclear translocation from the ER [13,31]. Human cancer cell lines, including HCT116, DLD1, and PANC1 cell lines, exhibit nuclear localization of NRF3, implying that these cancer cells undergo NRF3-activating stress. Elucidating the characteristics of stress and stress-driven mechanisms of NRF3 activation will further improve our understanding of the biological functions of NRF3 in cancer.

Finally, the normal physiological roles of NRF3 are still poorly understood. *Nrf3*-deficient mice display no apparent abnormalities, implying that the function of this gene is limited in specific cells or is compensated for by other transcription factors, especially NRF1. Nevertheless, we also believe that there are functional differences between NRF3 and NRF1 because cancer cells positively select NRF3 over NRF1 via CPEB3. I surmise that the specific function of NRF3 is exerted only in certain cell types or contexts under normal conditions and that dysregulation of *NRF3* expression leads to tumorigenesis and malignancy.

## Figures and Tables

**Figure 1 cancers-12-02681-f001:**
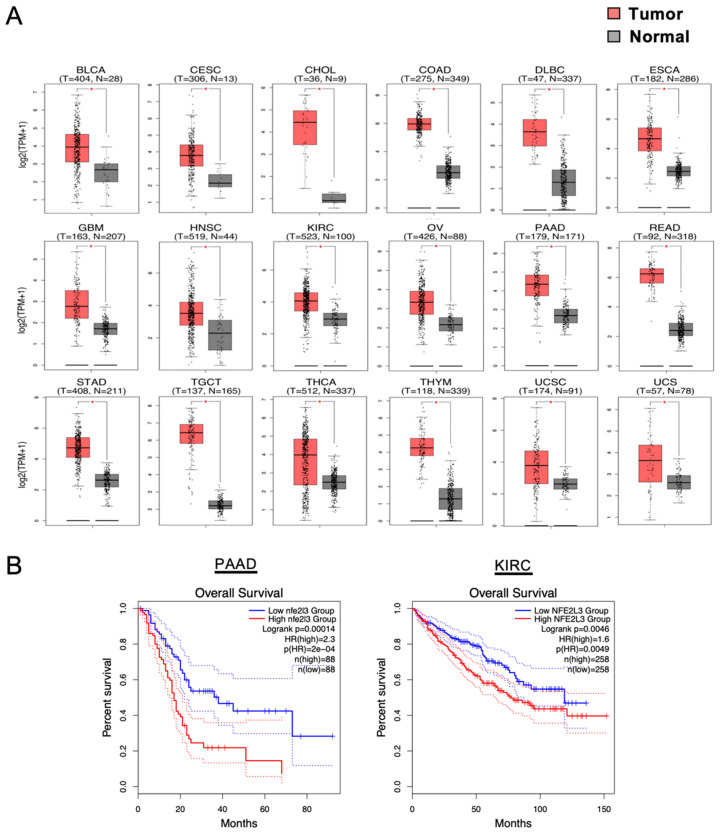
High upregulation of the NFE2-related factor 3 (NRF3) gene with malignancy. (**A**) Box plots depict the *NRF3* gene expression profiles across multiple cancer types and in paired normal samples from the Gene Expression Profiling Interactive Analysis 2 (GEPIA2) web server [16]. Each red and gray box represents a tumor and adjacent normal specimen, respectively. * *p* < 0.05. BLCA, bladder urothelial carcinoma; CESC, cervical squamous cell carcinoma and endocervical adenocarcinoma; CHOL, cholangiocarcinoma; COAD, colon adenocarcinoma; DLBC, diffuse large B-cell lymphoma; ESCA, esophageal carcinoma; GBM, glioblastoma multiforme; HNSC, head and neck squamous cell carcinoma; KIRC, kidney chromophobe; OV, ovarian serous cystadenocarcinoma; PAAD, pancreatic adenocarcinoma (it is also annotated as PDAC—pancreatic ductal adenocarcinoma); READ, rectum adenocarcinoma; STAD, stomach adenocarcinoma; TGCT, testicular germ cell tumors; THCA, thyroid carcinoma; THYM, thymoma; UCEC, uterine corpus endometrial carcinoma; UCS, uterine carcinosarcoma. (**B**) Kaplan–Meier analyses comparing overall survival between groups expressing higher and lower levels of NRF3. The hazard ratio (HR) was calculated on the basis of Cox’s proportional hazards model. Data are from patients with PAAD and KIRC from GEPIA2.

**Figure 2 cancers-12-02681-f002:**
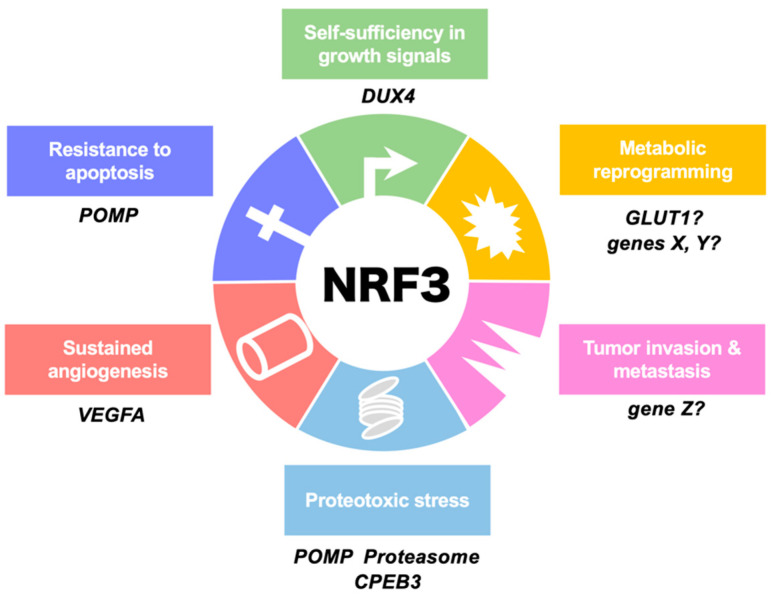
Involvement of NRF3 in the hallmarks of cancer. This illustration shows the six hallmark cancer capabilities originally proposed by Hanahan and Weinberg [1] and modified in Luo et al. [5]. NRF3 confers cells with these capabilities by inducing the expression of the indicated genes. We are currently examining biological functions of X-Z genes in each hallmark. POMP, proteasome maturation protein; DUX4, double homeobox 4; GLUT1, glucose transporter 1; CPEB3, cytoplasmic polyadenylation element (CPE)-binding protein 3; VEGFA, vascular endothelial growth factor A.

**Figure 3 cancers-12-02681-f003:**
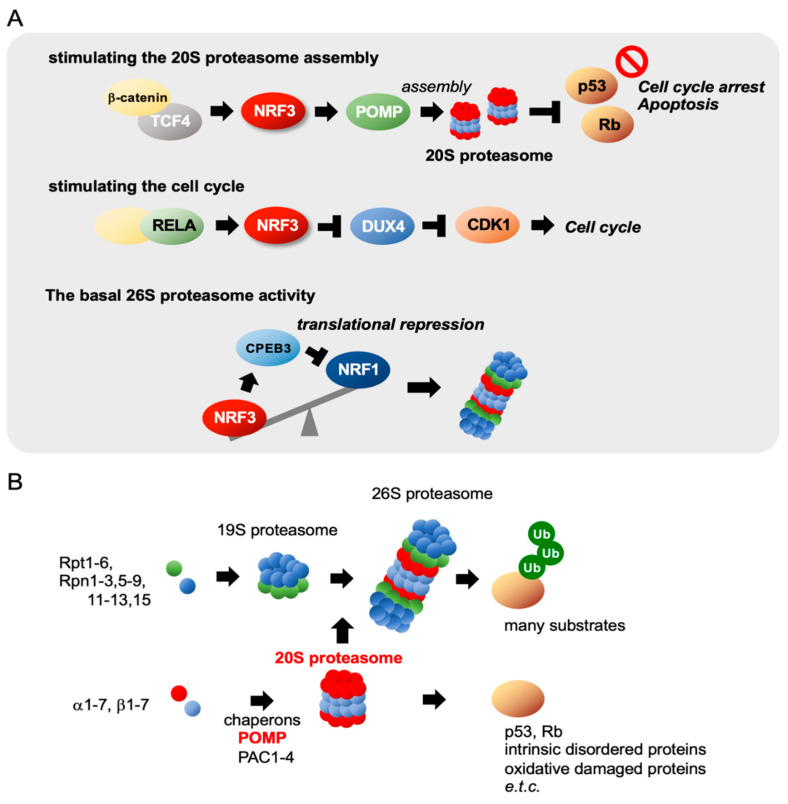
Three biological functions of NRF3 in cancer. (**A**) NRF3 exhibits three biological functions by regulating gene expression in colon cancer. (Top) In colorectal cancer cells, NRF3, the expression of which is induced by the β-catenin/T-cell transcription factor 4 (TCF4) complex, enhances the assembly of the 20S proteasome by inducing the proteasome maturation protein (POMP) chaperone gene. Consequently, the 20S proteasome degrades the tumor suppressors p53 and retinoblastoma (Rb) in a ubiquitin-independent manner, preventing cell cycle arrest and apoptosis. (Middle) NRF3 is also induced by the NF-kB complex, including RELA in colon cancer cells, and activates cell proliferation by repressing the CDK inhibitor DUX4 gene and consequently activating CDK1 [22]. (Bottom) NRF3 represses the translation of NRF1 by inducing cytoplasmic polyadenylation element-binding protein 3 (CPEB3) and modulates basal proteasome activity in colon cancer cells [27]. (**B**) Schematic representation of the molecular mechanisms of proteasome assembly. The 26S proteasome comprises 20S and 19S proteasomes containing 66 subunits [28,29,30]. The assembly of each proteasome complex is tightly regulated in a coordinated manner by distinct chaperones. The 20S proteasome is assembled by proteasome assembling chaperone 1–4 (PAC1–PAC4) and POMP chaperones.

**Figure 4 cancers-12-02681-f004:**
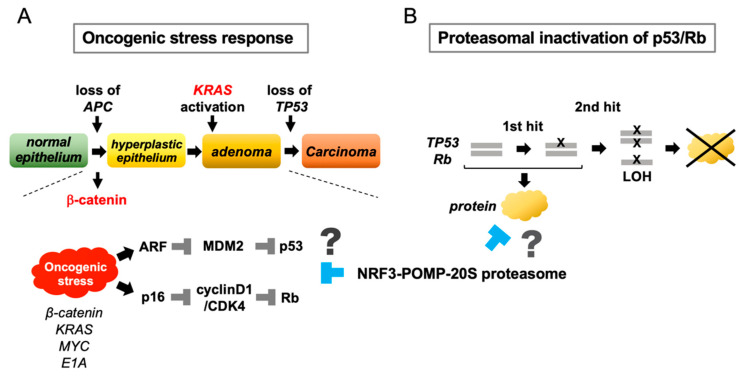
Conceptual hypotheses of the functional significance of NRF3 in cancer. The identification of the NRF3–POMP–20S proteasome axis allowed us to propose two conceptual hypotheses. (**A**) During the development of colorectal cancer, mutation of *APC* or *β-catenin* in the Wnt pathway is a crucial initial event that activates the oncogene β-catenin, and mutation of *TP53* is the final event in carcinogenesis. In normal cells, activation of individual oncogenes, namely, “oncogenic stress“, activates p53 and Rb by stimulating the tumor suppressors ARF and p16, leading to cell cycle arrest and apoptosis. This response of normal cells confers cytoprotection against oncogenic stress. The NRF3–POMP–20S proteasome axis may eliminate the oncogenic stress response in normal cells to enable tumorigenesis. (**B**) The NRF3–POMP–20S proteasome axis represses the biological functions of p53 and Rb via proteasomal degradation. The advantage of this mechanism may be that repression occurs without a two-hit mutation of these genes.

**Figure 5 cancers-12-02681-f005:**
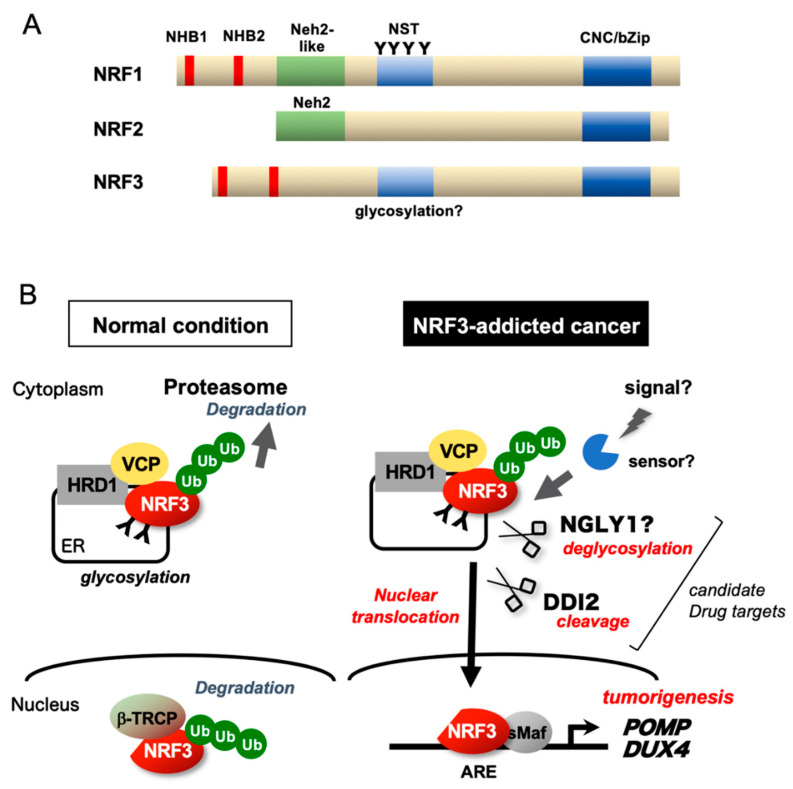
Regulatory mechanisms of NRF3. (**A**) Schematic structures of NRF1, NRF2, and NRF3. (**B**) Under physiological conditions, the transcriptional activity of NRF3 is suppressed by NRF3 sequestration in the endoplasmic reticulum (ER) via the N-terminal homology box 1 (NHB1) domain and HRD1-mediated proteasomal degradation. In NRF3-addicted cancer, NRF3 is liberated from multiple suppression mechanisms and processed at the NHB2 domain by the DNA damage-inducible 1 homolog 2 (DDI2) protease, thereby activating gene expression in the nucleus. Cap ‘n’ collar (CNC)-bZip, CNC-basic leucine zipper; β-TRCP, β-transducin repeat containing E3 ubiquitin protein ligase; ARE, antioxidant-response element; NST, Asn/Ser/Thr-rich; NHB, N-terminal homology box; sMaf, small Maf; Ub, ubiquitin; and VCP, valosin-containing protein.

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
