# Peer review of "Roles of NRF3 in the Hallmarks of Cancer: Proteasomal Inactivation of Tumor Suppressors"

_cancers, 2020, doi:10.3390/cancers12092681_

Round 1
Reviewer 1 Report
This paper reviews recent research work on understanding proteasomal functions of NRF3, a cancer hallmark transcription factor in many cancer types, especially in the colon and pancreatic cancers. The review was clearly written and well organized. It went through NRF3’s pathological roles along with clinical outcomes for multiple cancers and underlying mechanisms including gene regulation and proteostasis. Also, the author compared with other NRF family genes and TFs, and hypothesized that NRF3 has potential new roles in proteasomal inactivation of tumor suppression. Finally, the author envisioned medical translation such as drugs inducing NRF3 suppression. I just have a few minor comments.
(1) For gene regulation, the author mentioned no definitive conclusion due to a small number of ChIP peaks, but is it possible to relate NRF3 to the latest TCGA/ICGC epigenomic data (https://www.nature.com/articles/s41586-020-1969-6)?
(2) Can the author give more mechanistic insights on cross-talks between NRF3 and other TFs? Any protein-protein interactions or gene regulation? The author mentioned that NRF3 perhaps signals other master TFs. What are these master TFs and target genes?
Author Response
(1) For gene regulation, the author mentioned no definitive conclusion due to a small number of ChIP peaks, but is it possible to relate NRF3 to the latest TCGA/ICGC epigenomic data (https://www.nature.com/articles/s41586-020-1969-6)?
I appreciate the reviewer for providing me with important information about the TCGA/ICGC data. Unfortunately, I could not find the result of ChIP sequencing about NRF3 in the database. Instead, I found a dataset using human colon cancer LoVo cells (GSE51142) in the Gene Expression Omnibus (GEO). However, it shows also the small number of ChIP-peaks, and these peaks do not support our results and are therefore are unreliable. I consider that this issue is likely due to the authenticity of anti-NRF3 used in this experiment as described in section 8 in the manuscript.
(2) Can the author give more mechanistic insights on cross-talks between NRF3 and other TFs? Any protein-protein interactions or gene regulation? The author mentioned that NRF3 perhaps signals other master TFs. What are these master TFs and target genes?
I would like to thank the reviewer for the important suggestion about the lack of description in the manuscript. Because these data are preliminary, I would like to report these results in a next original paper after obtaining full data. It is likely that a protein interaction between NRF3 and the master transcription factor activates expression of target genes related to lipid metabolism. In addition, NRF3 likely induces expression of an EMT-related transcription factor, resulting in promoting cell migration.
Reviewer 2 Report
- Please correct some minor punctuation issues.
- The review is nicely written and it demonstrates the impact of a novel transcription factor, NRF3 in tumorigenesis. The only concern is that the introduction does not provide adequate background on this important transcription factor. Please incorporate that in this review.
Author Response
1. Please correct some minor punctuation issues.
I apologize for the grammatical errors in my manuscript. The manuscript has edited by native speakers again.
2. The review is nicely written and it demonstrates the impact of a novel transcription factor, NRF3 in tumorigenesis. The only concern is that the introduction does not provide adequate background on this important transcription factor. Please incorporate that in this review.
I would like to thank the reviewer for suggesting weakness in the manuscript and admit the criticism. According to the suggestion, I thought deeply about resolving the point, however, I could not succeed it. This review summarizes recent breakthroughs in our understanding of the physiological function of NRF3 that remained unclear for a long time and begins the introduction with a hypothesis about the physiological roles of NRF3 in tumorigenesis based on human cancer datasets. So it is difficult to incorporate further information in the introduction.